# Differences in Vascular Density between Detached and Nondetached Areas in Eyes with Rhegmatogenous Retinal Detachment

**DOI:** 10.3390/jcm11102881

**Published:** 2022-05-19

**Authors:** Mariko Sato, Takeshi Iwase

**Affiliations:** Department of Ophthalmology, Akita University Graduate School of Medicine, Akita 010-8543, Japan; mariko.s@med.akita-u.ac.jp

**Keywords:** choriocapillaris plexus, deep capillary plexus, optical coherence tomography angiography, rhegmatogenous retinal detachment, vessel density

## Abstract

We examined the vessel density (VD) of the deep capillary plexus (DCP) and choriocapillaris plexus (CCP) by optical coherence tomography (OCT) angiography in eyes with rhegmatogenous retinal detachment, which had similar amounts of detached and nondetached areas in the macula region, and then determined the morphology by OCT until 6 months after surgery. A total of 13 eyes of 13 patients whose average age was 55.8 ± 12.3 years and were successfully treated were enrolled in this study. Throughout the postoperative period, the VD of the DCP in the detached area decreased significantly compared to that in the nondetached area. Conversely, there was no significant difference in the VD of the CCP between the detached and the nondetached areas. The ratio of VD of both the DCP and CCP in the detached area to the in the nondetached area did not show significant changes during the follow-up period of 6 months. The ratio of VD of the DCP in the detached area to that in the nondetached area correlated significantly with the ratio of the external limiting membrane–ellipsoid zone (*r* = 0.57, *p* < 0.001) and ellipsoid zone–retinal pigment epithelium (*r* = 0.39, *p* < 0.001) thickness in the detached area to that in the nondetached area. A well-preserved DCP blood flow could result in the restoration of the outer retina.

## 1. Introduction

Rhegmatogenous retinal detachment (RRD) is a vision-threatening disease characterized by the separation of the neurosensory retina from the retinal pigment epithelium (RPE) with subretinal fluid accumulation through a retinal break(s) [1]. The situation causes outer retinal ischemia, with rapid and progressive photoreceptor degeneration [2]. Experiments conducted using animal and human models have demonstrated that photoreceptor cells die within 1–3 days after retinal detachment [3,4]. Various treatments have been used to attach the detached retina, including vitrectomy, scleral buckling, pneumatic retinopexy, or laser coagulation [5,6]. Despite the high anatomical restoration after surgery, the functional results show a wide range of visual outcomes [7]. The cause of suboptimal visual recovery after successful surgery is an extensively investigated problem.

Morphological studies conducting optical coherence tomography (OCT) have shown that damage to the outer retinal layer significantly affects vision in eyes with RRD. Kobayashi et al. retrospectively analyzed spectral domain (SD)-OCT images of eyes with successfully reattached macula-off RRD and found that the degree of restoration of the outer retinal layers at the fovea, especially the thickness of the ellipsoid zone (EZ)—RPE, is important in predicting postoperative visual acuity [8]. Moreover, the EZ/external limiting membrane (ELM) integrity [9] or the presence of a foveal bulge [10] were reported as predictors of visual acuity after macula-off RRD surgery.

The retinal microvasculature is an important component of the retinal structure and has been reported to be involved in the recovery in eyes with retinal detachment [11]. OCT angiography (OCTA) is a novel imaging system that allows us to observe each layer of retinal capillaries, e.g., superficial capillary plexus (SCP) and deep capillary plexus (DCP), and choriocapillaris plexus (CCP) blood flow in more detail in the macular region [11,12,13,14]. OCTA can provide further information on the changes occurring in the highly important process of macular vascularization after its reattachment in eyes with RRD. Recent studies using OCTA have demonstrated that the vessel density (VD) in the SCP and DCP [15], or in the CCP [12], was significantly decreased compared to that in the fellow eyes. McKay et al. reported that decreased VD in the DCP compared to the fellow healthy eyes was correlated with worse visual acuity [16]. It has been suggested that the vessel density in CCP could be related to the anatomical restoration of the outer retinal layer in macula-off RRD [17,18]. Thus, it is important to clarify the relationship between the VD and morphology after macula-off RRD surgery.

When evaluating OCTA images, understanding common artifacts is important to prevent any potential misinterpretation that may compromise clinical decision-making [17]. Most papers using OCTA images have strict criteria in place to eliminate poor-quality images and to ensure that the images included in the final analysis are of good quality. However, when media opacities, e.g., cataracts or dry eye are present, signal strength may be further degraded [17]. Even within the same image, the possibility that results may differ among locations due to artifacts, e.g., motion or shadowing artifacts, cannot be ruled out. There have been no reports so far comparing detached and nondetached areas in the same image, and we believe that comparing them in the same image has a certain significance in excluding the influence of artifacts caused by comparison with other eyes.

In this study, we investigated the VD of the DCP and CCP using OCTA in eyes with RRD that had similar amounts of detached and nondetached areas in the macular region. Our aim was to compare the VD in the DCP and CCP between detached and nondetached areas in addition to morphology until 6 months after surgery.

## 2. Patients and Methods

This was a retrospective, cross-sectional, single-center study. The Ethics Committee of Akita University Hospital (Akita, Japan) approved the procedures, and the procedures conformed to the tenets of the Declaration of Helsinki. Informed consent was obtained from all participants after explaining the nature and possible complications of the study.

### 2.1. Subjects and Testing Protocol

Patients were reviewed who had been diagnosed with primary RRD and surgically treated for RRD between April 2020 and August 2021 at the Department of Ophthalmology of Akita University Hospital. A total of 13 eyes of 13 patients were successfully treated with a single, uncomplicated surgical procedure and underwent SD-OCT and OCTA preoperatively, and at 2 weeks, 1, 2, 3, 4, 5, and 6 months postoperatively, were included in this study. All patients underwent a comprehensive ophthalmic examination, including measurements of best-corrected visual acuity (BCVA), intraocular pressure (IOP), axial length, slit-lamp examination, and fundus examination. Snellen VA values were converted into the logarithm of the minimum angle of resolution (LogMAR) units to generate a linear scale of VA. The VD of the DCP and CCP was measured in a 0.45 mm × 0.45 mm area centered at 1.2 mm from the fovea in a contralateral position to the fovea in the detached area and nondetached area. Only cases with complete detachment on one side of the measurement site and complete no detachment on the other side were enrolled in this study.

Exclusion criteria were a high myopia (axial length ≥ 27.5 mm), dense ocular media (e.g., vitreous hemorrhage, vitreous opacity), preexisting macular conditions (e.g., macular degeneration, vascular occlusive diseases, or diabetic retinopathy), proliferative vitreoretinopathy of ≥grade C [19], and clinically evident postoperative change likely to interfere with accurate evaluation of retinal layers (e.g., recurrent RRD, epiretinal membrane, cystoid macular edema, or persistent subretinal fluid 2 weeks after surgery). Patients were also excluded if their SS-OCT and OCTA measurements showed poor scan quality [20], measurement errors (such as segmentation error, centering error, or fixation error), or artifacts (defocus, blink lines, or motion artifacts).

### 2.2. Surgical Techniques

Retinal detachment surgery was performed with a conventional scleral buckling procedure or PPV. For the scleral buckling surgery, the retinal breaks were treated by transscleral cryotherapy. A silicone sponge for the circumferential segmental buckle was sutured as an explant in all cases. The SRF was drained if necessary.

For the PPV surgery, after creating the three ports, PPV was performed using a 25-gauge system. After the vitreous was removed, fluid-air exchange and subretinal fluid drainage and endo-photocoagulation were applied to the causative retinal tear(s). Then 20% sulfur hexafluoride (SF6) was injected into the vitreous upon completion of the PPV.

### 2.3. OCTA Imaging

A PLEX Elite^®^ OCTA (Carl Zeiss Meditec, Dublin, CA, USA) was used to obtain en face images of the microvasculature around the fovea. The PLEX Elite^®^ is a swept-source (SS)-OCTA instrument that uses a swept laser source with a central wavelength of 1040 to 1060 nm (980–1120 nm full bandwidth) and operates at 100,000 A-scans per second. The eye with RRD was imaged using the 3 × 3 mm scan protocols centered on the fovea. All scans were reviewed by two graders (MS and TI). The VD of the DCP and CCP was measured in a 0.45 × 0.45 mm area centered at 1.2 mm from the fovea in a contralateral position to the fovea in the detached area and nondetached area (Figure 1). OCTA images of the DCP and CCP were binarized by the Otsu method [21]. The VD was calculated as the ratio of the area occupied by vessels divided by the total area of the binarized image. The ratio of VD ([detached area]/[nondetached area]) of the DCP and CCP was measured.

### 2.4. SD-OCT Imaging

A Spectralis^®^ SD-OCT (Heidelberg Engineering, Heidelberg, Germany) was used to obtain all SD-OCT images. We evaluated the recorded vertical cross-section images. The retinal layer thickness was measured on the same selected central foveal scan using the computer-based caliper measurement tool of the SD-OCT system [8]. The ELM-EZ thickness was defined as the distance between the outer borders of the ELM and EZ. The EZ-RPE thickness was defined as the distance between the outer border of the EZ and the inner border of the RPE. These thicknesses were measured at the same locations as the OCTA measurements, i.e., 1.2 mm above and below the fovea. The ratio of thickness ([detached area]/[nondetached area]) of the ELM-EZ and EZ-RPE was measured.

### 2.5. Statistical Analyses

All statistical analyses were conducted by using IBM SPSS Statistics for Windows, Version 26.0 (IBM Corp., Armonk, NY, USA). Data are presented as the mean ± standard deviation. Kolmogorov–Smirnov test was used to evaluate the normality of sample distribution. Mann–Whitney U test was used for two non-parametric independent samples. A mixed linear model was used to evaluate changes in the ratio of the VD in the DCP and CCP, and of the thickness of the ELM-EZ and EZ-RPE. Spearman’s correlation coefficient was used for non -parametric correlations. A *p*-value of <0.05 was considered significant.

## 3. Results

### 3.1. Clinical Characteristics of the Subjects

We enrolled 13 patients (10 men and 3 women) with an average age of 55.8 ± 12.3 years (range 37–72 years) in this study. They complained of decreased vision or visual field defects for 9.1 ± 10.1 days (range 1–30 days). The mean axial length was 25.4 ± 1.6 mm (range 23.4–27.4 mm) in the affected eyes. The mean logMAR BCVA was 0.33 ± 0.44 (range −0.07–1.22) before surgery and significantly improved to −0.01 ± 0.13 (range −0.07–0.39) 6 months after surgery (*p* = 0.007, Wilcoxon signed-rank test). The mean IOP was 14.1 ± 2.3 mmHg (range 11–17.0 mmHg) before surgery. Of the affected eyes, 11 were treated with PPV and 2 were treated with the segmental buckling procedure. Table 1 shows the clinical characteristics of the subjects.

### 3.2. Comparison of VD in the DCP and CCP between the Detached Area and Nondetached Areas

The VD of the DCP in the detached area significantly decreased compared to that in the nondetached area (2 W; *p* = 0.038, 1 M; *p* = 0.036, 6 M; *p* = 0.043, Mann–Whitney U test) (Figure 2). The ratio of the VD of the DCP in the detached area to that in the nondetached area was not significantly different during the follow-up period. The VD of the CCP was not significantly different between the detached and nondetached areas. Furthermore, the ratio of the VD of the CCP in the detached area to that in the nondetached area was not significantly different during the follow-up period.

### 3.3. Comparison of ELM-EZ and EZ-RPE Thickness between Detached Area and Nondetached Areas

The ELM-EZ thickness in the detached area was significantly lesser than that in the nondetached area (2W; *p* = 0.041, 1M; *p* = 0.049, Mann–Whitney U test) (Figure 3). The EZ-RPE thickness in the detached area was also significantly lesser than that in the nondetached area (2W; *p* = 0.002, 1M; *p* = 0.003, 2M; *p* = 0.011, 3M; *p* = 0.013, 4M; *p* = 0.019, Mann–Whitney U test). The ELM-EZ thickness in the detached area tended to elongate during the follow-up period, but it was not significantly different. In contrast, the ratio of the VD in the ELM-EZ thickness of the detached area to that of the nondetached area increased significantly during the follow-up period (*p* < 0.001).

### 3.4. Correlation of VD and Retinal Thickness between Detached Area and Nondetached Areas

The ratio of the VD of the DCP in the detached area to that in the nondetached area correlated significantly with the ratio of the ELM-EZ thickness in the detached area to that in the nondetached area (*r* = 0.47, *p* < 0.001) (Figure 4). Moreover, the ratio of the VD of the DCP in the detached area to that in the nondetached area correlated significantly with the ratio of the EZ-RPE thickness in the detached area to that in the nondetached area (*r* = 0.33, *p* < 0.001) (Figure 5). However, the ratio of the VD of the CCP in the detached area to that in the nondetached area did not correlate significantly with the ratio of both ELM-EZ thickness and EZ-RPE thickness in the detached area to that in the nondetached area.

## 4. Discussion

The VD of the DCP in the detached area decreased significantly compared to that in the nondetached area. However, there was no significant difference in the VD of the CCP between the detached and nondetached areas. The ratio of the VD of both DCP and CCP in the detached area to that in the nondetached area showed no significant changes during the follow-up period of 6 months. The ratio of the VD of the DCP in the detached area to that in the nondetached area correlated significantly with the ratio of ELM-EZ and EZ-RPE thicknesses in the detached area to that in the nondetached area.

OCTA is a novel noninvasive method to visualize and provide quantitative vascular information on macular microcirculation, including the retinal capillary plexus and CCP with good repeatability and reproducibility [11,12,13,14]. Studies have reported that the VD in the SCP, DCP, and CCP decreased significantly compared with the fellow eyes or macula-off eyes [12,15,22]. However, a comparison in different OCTA images may affect the results [23,24]. Therefore, it is better to compare the detached and nondetached areas within the same image to obtain more accurate results. Moreover, there is limited knowledge on the changes in VD over time for as long as 6 months after surgery. Hence, we compared the VD of the DCP and CCP between the detached and nondetached areas in the macula using the same image with similar amounts of detached and nondetached areas until 6 months after surgery.

We observed a significant decrease in the VD of the DCP in the detached area compared to that in the nondetached area throughout the postoperative period. This result is in good agreement with previous reports [13,15,25], which have documented impaired retinal microcirculation in the macula of eyes with RRD and reduced VD of the microvasculature compared with other eyes and control eyes. In eyes with RRD, the detached retina becomes hypoxic because of a lack of blood supply from the choroid, the retinal capillaries become dilated and permeable, and diffuse vascular occlusions are observed [26,27,28].

The ratio of the VD of the DCP in the detached area to that in the nondetached area was not significantly different after surgery. This result clearly indicates that the VD of the detached area does not change in the short postoperative period of 6 months. It also implies that it may take a long time for blood perfusion to be restored. However, Wang et al. reported that the parafoveal VD of the DCP in the detached eyes was lower than that in the fellow eye 2 and 4 weeks after surgery, but it significantly increased over time at 12 weeks of follow-up. It has also been reported that the retinal blood flow to the optic nerve head was significantly reduced in eyes with preoperative RRD but can recover after successful RRD repair with PPV [29]. In contrast, Resch et al. reported that the superficial foveal region and the extent of the nonflow area are more involved after a longer postoperative period, although the deep and parafoveal region is more affected in the earlier postoperative period [13]. The reason for the discrepancy between our result and their result is unclear, but we compared the VD between the detached and nondetached area within the same OCTA image. Thus, our result should not be affected by the quality of the OCTA image. However, there may be cases where the OCTA image quality is not good because of the opacity of ocular media, e.g., inflammation and vitreous hemorrhage in the early postoperative period. Moreover, there is a possibility that these conditions will disappear with time and the image quality will become clearer, resulting in an increase in VD.

In this study, the EZ–RPE thickness measured at the same site as the VD measurement in the detached area was shorter than that in the nondetached area early after successful retinal attachment and significantly increased over time. Several previous studies have reported the recovery of the outer retinal thickness at the fovea and photoreceptors, over time, after retinal detachment surgery in eyes with macula-off RRD [8,30]. Our result is consistent with these reports. After retinal detachment, nutrition to the photoreceptor cells is impaired and the photoreceptor cells begin to degenerate rapidly through apoptosis and programmed cell death [31]. Moreover, progressive recovery of thickness and restoration of the outer retinal layers/bands at the fovea occur following macula-off RRD repair [8].

We further analyzed the correlation between the OCT and OCTA parameters. Our results showed that the ratio of the VD in the detached area to that in the nondetached area correlated significantly with the ratio of ELM–EZ and EZ–RPE thicknesses after successful retinal attachment. This result suggests that the VD of the DCP is related to the morphological restoration of the outer retina. Hence, a well-preserved DCP blood flow could result in the restoration of the outer retina.

The VD of the CCP in the detached area did not significantly decrease compared to that in the nondetached area throughout the postoperative period. In addition, the ratio of the VD of the CCP in the detached area to that in the nondetached area did not correlate with the ratio of ELM–EZ and EZ–RPE thicknesses after surgery. This result was not consistent with previous reports, which have demonstrated impaired microcirculation in the CCP in eyes with macula-off RRD and reduced VD of the microvasculature compared with other eyes and control eyes [12,18]. However, the causal relationship between the decreased CCP blood flow and the defect of the photoreceptor layer in eyes with macula-off RRD could not be elucidated, although the areas of flow voids in the CCP layer beneath the area of disrupted photoreceptors was less than that in the CCP layer beneath the area of intact photoreceptors in ischemic maculopathy [32]. After the occurrence of RRD, the outer layer of the detached retina shortens, but this may not affect the blood flow of the CCP after successful attachment.

McKay et al. reported that decreased VD in the DCP compared to the fellow healthy eyes was correlated with worse visual acuity [16]. On the other hand, we did not evaluate the relationship between visual acuity and VD. We focused on comparing the VD and the morphology in the DCP and CCP between detached and nondetached areas that did not include the fovea. It would be clinically relevant and interesting to determine which of all the measures at baseline might have predictive value for the final VA, but it was difficult to examine them in this study because the area did not include the fovea.

This study has some limitations. First, this was a retrospective study with a relatively small sample size, which may have resulted in an insufficient number of participants for adequate comparison. However, it is not easy to collect these cases because of the scarcity of cases in which the detached and nondetached areas are similar in the macular region. Second, eyes with RRD were treated with different surgical techniques, i.e., PPV and scleral buckling surgery. However, we excluded eyes with residual subretinal fluid 2 weeks after surgery from this study. Therefore, it should not have a significant impact on the postoperative VD between eyes treated with PPV and those treated with scleral buckling surgery. Third, we did not measure the VD in the SCP. As PLEX Elite 9000 does not have a function to exclude retinal blood vessels contained in the SCP slab images, the measurement of VD is affected by retinal blood vessels. Hence, we could not measure the VD in the SCP. Fourth, we did not measure the thickness of the inner retinal layer. Nevertheless, several reports have shown that the thickness of the inner retinal layer does not differ significantly between eyes with RD and fellow eyes after surgery [12,30]. Furthermore, the duration of macula-off RRD before surgery was relatively short in our study. In an experimental study conducted by Guerin et al., the inner segment of the retina remained intact for 7 days after retinal detachment [33]. Further prospective studies on a larger number of eyes with more detailed information on VD changes, including the fovea, morphology, and visual acuity, are required to confirm our findings determined by OCTA.

## 5. Conclusions

In conclusion, the VD of the DCP in the detached area decreased significantly and correlated with morphological restoration after successful retinal attachment. A well-preserved DCP blood flow could result in the restoration of the outer retina.

## Figures and Tables

**Figure 1 jcm-11-02881-f001:**
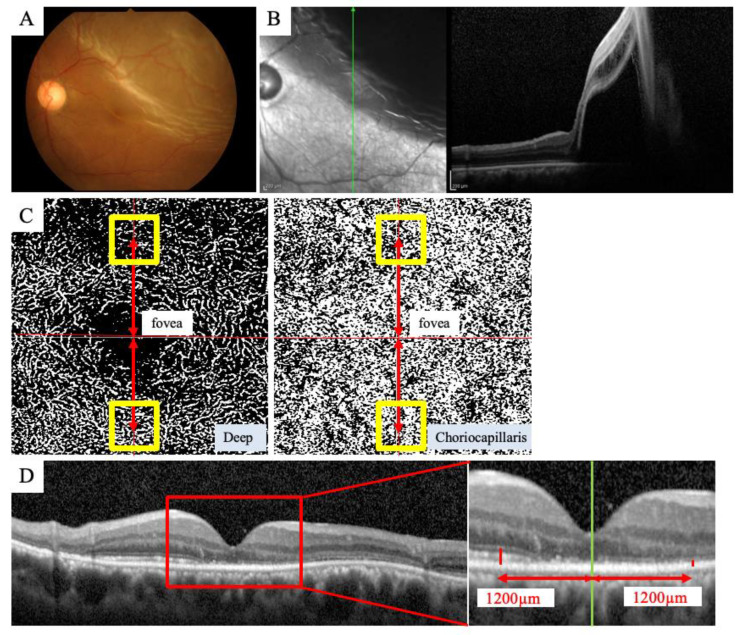
Measurements of VD using OCTA and outer retinal thickness using OCT in representative macula-off RRD, which had similar amounts of detached and nondetached areas in the macular region. Fundus photograph showed RRD (**A**), and OCT showed that the fovea has detached from the retina, resulting in similar amounts of detached and nondetached areas in the macular region (**B**). OCTA image of 3 × 3-mm scan protocols centered on the fovea was used for the measurement of VD (**C**). The VD of the DCP and CCP was measured in a 0.45 × 0.45-mm area centered at 1.2 mm from the fovea in a contralateral position to the fovea in the detached and nondetached areas. Outer retinal thicknesses were measured on the same selected vertical central foveal scan at the same locations as the OCTA measurements that were 1.2 mm above and below the fovea (**D**).

**Figure 2 jcm-11-02881-f002:**
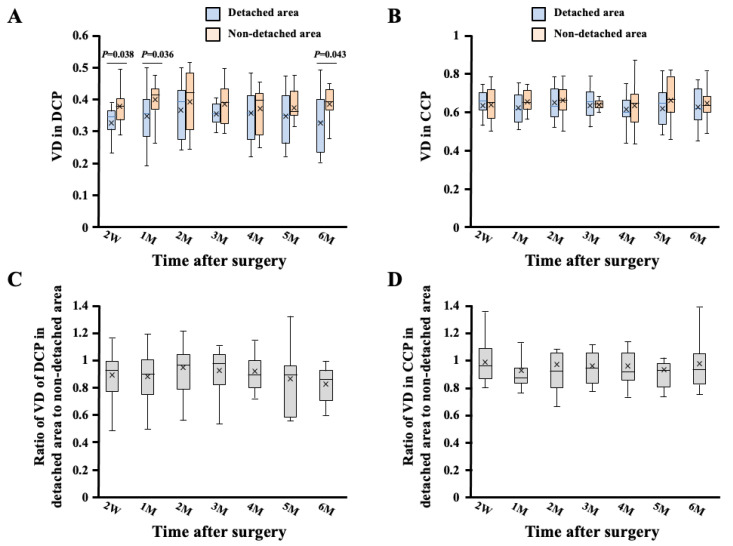
Box-and-whisker plot of comparisons of the VD of the DCP and CCP between detached and nondetached areas. The bottom and top of the box are the first and third quartiles, and the band inside the box is the median. X mark is the mean. The ends of the whiskers represent the minimum and maximum of all the data. The VD of the DCP in the detached area decreased significantly compared to that in the nondetached area (2W; *p* = 0.038, 1M; *p* = 0.036, 6M; *p* = 0.043, Mann–Whitney U test) (**A**). The ratio of the VD of the CCP in the detached area to that in the nondetached area was not significantly different during the follow-up period (**B**). The ratio of the VD of the DCP (**C**) and CCP (**D**) in the detached area to that in the nondetached area was not significantly different during the follow-up period.

**Figure 3 jcm-11-02881-f003:**
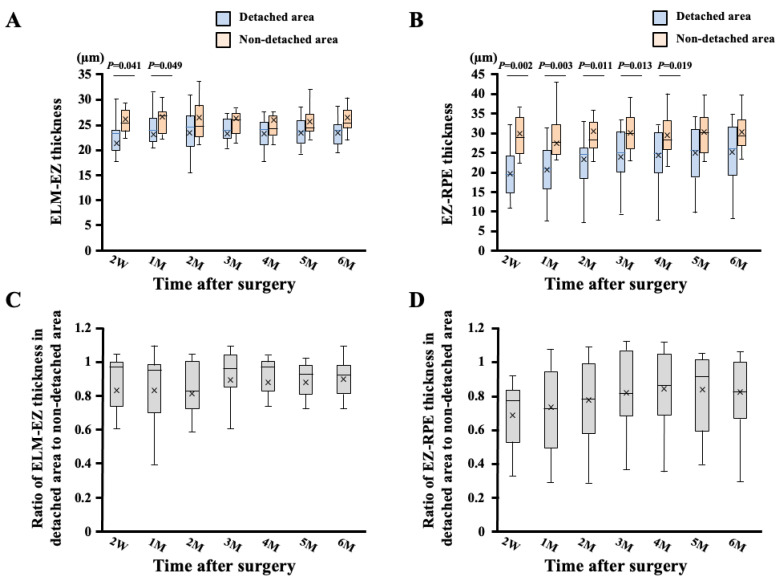
Box-and-whisker plot of comparisons of ELM-EZ and EZ-RPE thicknesses between detached and nondetached areas. The ELM-EZ thickness in the detached area was significantly lesser than that in the nondetached area (2W; *p* = 0.041, 1M; *p* = 0.049, Mann–Whitney U test) (**A**). The EZ-RPE thickness in the detached area was significantly lesser than that in the nondetached area (2W; *p* = 0.002, 1M; *p* = 0.003, 2M; *p* = 0.011, 3M; *p* = 0.013, 4M; *p* = 0.019, Mann–Whitney U test) (**B**). The ratio of the ELM-EZ thickness in the detached area to that in the nondetached area was not significantly different during the follow-up period (**C**). The ratio of the EZ-RPE thickness in the detached area to that in the nondetached area increased significantly during the follow-up period (*p* < 0.001) (**D**).

**Figure 4 jcm-11-02881-f004:**
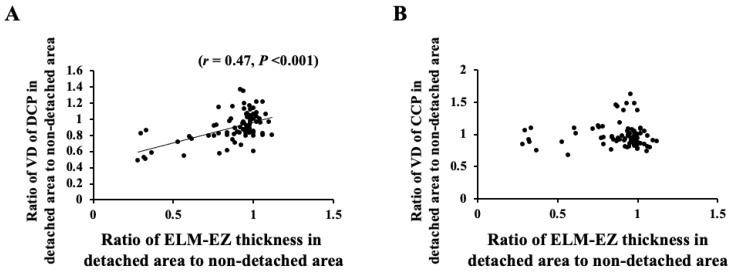
The ratio of the VD of the DCP in the detached area to that in the nondetached area correlated significantly with the ratio of the ELM-EZ thickness in the detached area to that in the nondetached area (*r* = 0.47, *p* < 0.001) (**A**), but that of the CCP did not correlate significantly with the ratio of ELM-EZ thickness (**B**).

**Figure 5 jcm-11-02881-f005:**
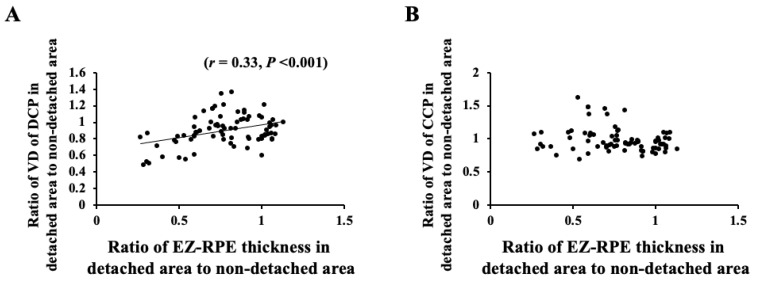
The ratio of the VD of the DCP in the detached area to that in the nondetached area correlated significantly with the ratio of the EZ-RPE thickness in the detached area to that in the nondetached area (*r* = 0.33, *p* < 0.001) (**A**), but that of the CCP did not correlate significantly with the ratio of ELM-EZ thickness (**B**).

**Table 1 jcm-11-02881-t001:** Clinical characteristics of the RRD eyes.

Clinical Characteristics of the RRD Eyes	
Age (y)	55.8 ± 12.3
Sex (male/female)	10/3
Duration between symptoms and surgery (days)	9.1 ± 10.1
Axial length (mm)	25.4 ± 1.6
Preoperative IOP (mmHg)	14.2 ± 2.2
Preoperative BCVA (Log MAR)	0.33 ± 0.44
Preoperative BCVA (Log MAR)	0.01 ± 0.14
Surgical procedure (PPV/buckle)	11/2

## Data Availability

The data that support the findings of this study are available from the corresponding author upon reasonable request.

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
