# Peer review of "Differences in Vascular Density between Detached and Nondetached Areas in Eyes with Rhegmatogenous Retinal Detachment"

_jcm, 2022, doi:10.3390/jcm11102881_

Round 1
Reviewer 1 Report
This is a review about the treatment in Choroidal Haemangioma.
This is not a novelty but it's important to have an updated resume about this issue.
This is relevant and original because there are not so many reports about this theme.
This is an updated review, including the most recent treatments as anti-vegf. It is well-written and clear. Reading is easy.The conclusions are a resume about the most interesting results.
There is no big novelty in this area and choroidal haemangiomas are not so frequent that science is always in update. But the last one have more than 10 years.
This paper can be interesting but results are disppointing but the low number of patients in the sample is probably the main reason for that. English can be imprroved and figures can be better explained.
Author Response
We are sorry, but these comments do not seem to be the same as the comments on our manuscript. Our paper is about the vascular density after rhegmatogenousretinal detachment, not about the treatment in choroidal haemangioma.
Reviewer 2 Report
Introduction:
Line 40 is incomplete
Lines 52-54: ‘’However, although several reports have shown decreased VD in eyes wtih RRD, the VD in eyes with RRD was compared with the fellow eye or control eyes.’’ The authors present the fact that microvascular metrics in RRD eyes were compared to fellow eyes as a weak study design, whereas it’s probably the strongest study design there is as confounding factors often present when comparing eyes of different subjects are eliminated.
Lines 54-55: ‘’Comparison in different OCTA images, especially in images of not excellent quality, may impact the results’’ The authors assume that studies comparing OCTA microvascular metrics in RRD vs fellow eye or vs control eye included images of potentially poor quality. This is probably not the case as in the vast majority of OCT-A articles that are published, authors are very meticulous in excluding images of poor quality and setting strict inclusion criteria to make sure that images that are included in the final analysis are of good quality.
Lines 55-57: It seems that the authors have written lines 52-55 just to attempt to artificially create the need for the present study that compares microvascular metics in detached and non-detached ares within the same OCTA image. ‘‘Therefore, we believe that it is better to compare the detached and nondetached areas 56 within the same image to obtain more accurate results’’ Have the authors considered the possibility that artifacts may as well exist in different areas within the same image ? It is certainly not uncommon for some areas within the same OCTA image to be of good quality and other areas of port quality due to motion artifacts, shadowing or any other kind of artifact. This would compromise this study’s claimed strength the same way that the authors claim that comparing RD eyes to fellow eyes may result in comparing images of different quality ie comparing areas of different quality within the same image.
The cornerstone swept source OCT/OCTA article doi: 10.1016/j.preteyeres.2021.100951. by Laíns et al must be mentioned somewhere in this work’s introduction.
The early swept source OCTA work on RD eyes vs fellow eyes by McKay et al doi: 10.2147/OPTH.S214623 is also missing in the introduction and discussion sections, important to mention as they found that decreased VD in the DCP in the RD eyes vs fellow eyes correlated with worse VA.
Methods:
Lines 88-89: It has been shown that PPV vs buckle affect VD in a different way. Although the authors mention in their limitations that different techniques were used, they down mention that it has been shown that they confound VD metrics. Maybe better exclude the 2 eyes that underwent a buckle.
Shouldn't ‘similar amounts of detached and nondetached areas in the macular region.’ be an inclusion criterion based on the introduction ? How did the authors determine that ?
Results:
It would be clinically relevant and interesting to investigate which of all the metrics that the authors measured at baseline could have a predictive value for final VA (baseline predictors for final VA at 6 months)
Discussion:
Well written
Limitations: VD in the SCP is has been previously reported in RD eyes. Not sure why the authors did not add this to the present work
Author Response
Line 40 is incomplete
Answer: We have revised the sentences.
Lines 52-54: ‘’However, although several reports have shown decreased VD in eyes wtih RRD, the VD in eyes with RRD was compared with the fellow eye or control eyes.’’ The authors present the fact that microvascular metrics in RRD eyes were compared to fellow eyes as a weak study design, whereas it’s probably the strongest study design there is as confounding factors often present when comparing eyes of different subjects are eliminated.
Answer: We agree with the reviewer’s comment. We have modified the sentences in line 58-60.
Lines 54-55: ‘’Comparison in different OCTA images, especially in images of not excellent quality, may impact the results’’ The authors assume that studies comparing OCTA microvascular metrics in RRD vs fellow eye or vs control eye included images of potentially poor quality. This is probably not the case as in the vast majority of OCT-A articles that are published, authors are very meticulous in excluding images of poor quality and setting strict inclusion criteria to make sure that images that are included in the final analysis are of good quality.
Answer: We agree with the reviewer’s comment which authors are very meticulous in excluding images of poor quality and setting strict inclusion criteria to make sure that images that are included in the final analysis are of good quality. We have modified the sentences in line 60-63.
Lines 55-57: It seems that the authors have written lines 52-55 just to attempt to artificially create the need for the present study that compares microvascular metics in detached and non-detached areas within the same OCTA image. ‘‘Therefore, we believe that it is better to compare the detached and nondetached areas 56 within the same image to obtain more accurate results’’ Have the authors considered the possibility that artifacts may as well exist in different areas within the same image ? It is certainly not uncommon for some areas within the same OCTA image to be of good quality and other areas of port quality due to motion artifacts, shadowing or any other kind of artifact. This would compromise this study’s claimed strength the same way that the authors claim that comparing RD eyes to fellow eyes may result in comparing images of different quality ie comparing areas of different quality within the same image.
Answer: We thanks to the reviewer for pointing this out and agree with the comment. We have modified the sentences in line 63-67.
The cornerstone swept source OCT/OCTA article doi: 10.1016/j.preteyeres.2021.100951. by Laíns et al must be mentioned somewhere in this work’s introduction.
Answer: We have cited this reference as citation number 17, and have added the sentences in line 54-55, 58-63.
The early swept source OCTA work on RD eyes vs fellow eyes by McKay et al doi: 10.2147/OPTH.S214623 is also missing in the introduction and discussion sections, important to mention as they found that decreased VD in the DCP in the RD eyes vs fellow eyes correlated with worse VA.
Answer: We have cited this reference as citation number 16 and have added sentences in the introduction in line 52-54 and discussion sections in line 293-294.
Methods:
Lines 88-89: It has been shown that PPV vs buckle affect VD in a different way. Although the authors mention in their limitations that different techniques were used, they down mention that it has been shown that they confound VD metrics. Maybe better exclude the 2 eyes that underwent a buckle.
Answer: As the reviewer pointed out, different techniques were used in this study. However, since we excluded eyes which persistent subretinal fluid remained at 2 weeks after surgery, all included eyes in this study had absorbed the subretinal fluid within 2 weeks. Therefore, we did not exclude the two eyes in which buckling was performed, considering that the difference in technique did not affect the results.
Shouldn't ‘similar amounts of detached and nondetached areas in the macular region.’ be an inclusion criterion based on the introduction ? How did the authors determine that ?
Answer: Since the VD of the DCP and CCP was measured in a 0.45 × 0.45-mm area centered at 1.2 mm from the fovea in a contralateral position to the fovea in the detached area and nondetached area. Only cases with complete detachment on one side of the measurement site and complete no detachment on the other side were enrolled in this study. As a result, at least within 3mm × 3mm centered on the fovea, only those with approximately the same extent of detached and nondetached were included. We added the sentences in line 87-91.
Results:
It would be clinically relevant and interesting to investigate which of all the metrics that the authors measured at baseline could have a predictive value for final VA (baseline predictors for final VA at 6 months)
Answer: We agree with the reviewer’s comment. We focused on comparing the VD and the morphology in the DCP and CCP between detached and nondetached areas where did not include fovea. Certainly, it would be clinically relevant and interesting to determine which of all the measures at baseline might have predictive value for the final VA, but it was difficult to examine them in this study because the measured area did not include the fovea. We have added the sentences in line 294-299.
Discussion:
Well written
Answer: We greatly appreciate this comment.
Limitations: VD in the SCP is has been previously reported in RD eyes. Not sure why the authors did not add this to the present work
Answer: We thanks to the reviewer for pointing this out. PLEX Elite 9000 does not have a function to exclude retinal blood vessels contained in the SCP slab images, which would result that the measurement of VD is affected by retinal blood vessels. In other words, it cannot measure only the density of retinal capillaries in the SCP slab images, and the results will be highly dependent on the amount of retinal blood vessels present in the area. Hence, we could not measure the VD in the SCP. We have added the sentences in line 308-311.
Round 2
Reviewer 1 Report
Differences in vascular density between detached and nonde- 2 tached areas in eyes with rhegmatogenous retinal detachment
This is a paper that studies the difference between VD of DCP and CCP by OCT-A in attached and detached retina. There is a low number of patients involved and this is the most negative point of the study.
I think that there is no need for so much descriptive surgical techniques in method sections. If you can resume it, I think that would be ok.
In statistical analysis, I don’t think that 13 patients are a enough sample to perform parametric tests. I would like to see your normality distribution and if it is a non-normal sample, please change it for non-parametric tests.
In results section when you describe “… is significantly less…”, are you sure that is not “… is significantly inferior…” or “lesser” as a comparative adjective?
In figure 4, the correlation diagrams, can you please show the relationship between VD to ELM-EZ and EZ-RPE in detached ate non-detached separately?
Author Response
Differences in vascular density between detached and nondetached areas in eyes with rhegmatogenous retinal detachment.
This is a paper that studies the difference between VD of DCP and CCP by OCT-A in attached and detached retina. There is a low number of patients involved and this is the most negative point of the study.
I think that there is no need for so much descriptive surgical techniques in method sections. If you can resume it, I think that would be ok.
Answer: We have shortened the description of the surgical techniques in method sections.
In statistical analysis, I don’t think that 13 patients are a enough sample to perform parametric tests. I would like to see your normality distribution and if it is a non-normal sample, please change it for non-parametric tests.
Answer: We have evaluated the normality of sample distribution with Kolmogorov–Smirnov test. Since it was a non-normal sample, we have changed it for non-parametric tests.
In results section when you describe “… is significantly less…”, are you sure that is not “… is significantly inferior…” or “lesser” as a comparative adjective?
Answer: We thank the reviewer for pointing out it. We have corrected the error to “lesser”.
In figure 4, the correlation diagrams, can you please show the relationship between VD to ELM-EZ and EZ-RPE in detached ate non-detached separately?
Answer: We have shown the relationship between VD to ELM-EZ and EZ-RPE separately as Figures 4 and 5.
Round 3
Reviewer 1 Report
Congratulations, you improved your work
In a non-normality distribution, you don’t use Mean ± SD, you can use median (IQR) or (min-max). To evaluate changes, why do you use a mixed model?
Author Response
Congratulations, you improved your work
In a non-normality distribution, you don’t use Mean ± SD, you can use median (IQR) or (min-max).
Answer: We thank the reviewer for pointing out it. Figure 2 and 3 have been modified to use Box-and-whisker plot.
To evaluate changes, why do you use a mixed model?
Answer: Since there were a few periods for which no data were available, we used a mixed model.